# Factors Influencing Quality of Life in Breast Cancer Patients Six Months after the Completion of Chemotherapy

**DOI:** 10.3390/diseases7010026

**Published:** 2019-02-24

**Authors:** Maria Lavdaniti, Dimitra Anna Owens, Polixeni Liamopoulou, Kalliopi Marmara, Efrosini Zioga, Michael S. Mantzanas, Eftychia Evangelidou, Eugenia Vlachou

**Affiliations:** 1Nursing Department, Alexander Technological Educational Institute, 57400 Thessaloniki, Greece; 2Psychologist, MSc, 19009 Athens, Greece; demikitty@hotmail.com; 3Nursing Department, Alexander Technological Educational Institute, 57400 Thessaloniki, Greece; tzeni_liam@yahoo.com; 4Faculty of Physical Education, Aristotele University of Thessaloniki, 54124 Thermi Thessaloniki, Greece; popimarmara@hotmail.co.uk; 5General Hospital Imathias, 59132 Veria, Greece; efr.zioga@gmail.com; 6Department of University Orthopedic, MSc, MHSc, PhD, General Hospital of Nea Ionia “Konstantopouleio-Patision”, 14233 Athens, Greece; msmantzanas@yahoo.gr; 7Department of Infection Control, MSc, MHSc, PhD(c), General Hospital of Nea Ionia “Konstantopouleio-Patision”, 14233 Athens, Greece; efievaggelidou@yahoo.gr; 8Nursing Department, University of West Attica, 12243 Athens, Greece; evlachou@uniwa.gr

**Keywords:** quality of life, breast cancer, chemotherapy

## Abstract

Purpose: To assess breast cancer patients’ quality of life six months after the completion of adjuvant chemotherapy, and to investigate factors affecting this. Methods: The study was conducted in one large hospital located in a major Greek city. A convenience sample of 61 breast cancer outpatients was recruited. A questionnaire, including the SF-36 scale and questions regarding demographic and clinical information, was used to collect data. Results: The mean age of the patients was 51.52 ± 12.10. The effect of age on the physical role was significant (*p* = 0.003). Τhe effect of menopausal status on physical role was also found to be significant (*p* = 0.003); this might be explained by age. Regarding the treatment type, patients who received hormone therapy in addition to surgery and chemotherapy reported a significantly higher quality of life in terms of bodily pain (*p* = 0.04) and vitality (*p* = 0.04) than patients who underwent only surgery and chemotherapy. Conclusions: Quality of life is affected by factors such as age, menopausal status, and previous therapy. Health care professionals should be more aware of the factors that influence the quality of life domains (physical role, bodily pain, vitality) within this group of cancer patients in order to meet their needs following acute treatment.

## 1. Introduction

Breast cancer is the most common cancer in women worldwide. In Greece, it is estimated that one in twelve women will be affected by breast cancer during her lifespan. Almost 4500 women are diagnosed with breast cancer every year [1]. It is the leading cancer type in women in all European countries [2].

Quality of life (QOL) is defined as “an individual’s perception of his/her position in life in the context of the culture and value systems in which he/she lives and in relation to his/her goals, expectations, standards, and concerns” by the World Health Organization Quality of Life Group. This definition considers that quality of life is a subjective concept and depends on an individual’s judgment [3].

Breast cancer and its treatment impacts patients’ quality of life [4]. Quality of life in breast cancer patients has received increasing attention due to a rising number of new cases, an improved survival rate, and the vital role of the woman in the family [5].

Several studies have examined the determinants of quality of life in breast cancer survivors and in patients after completion of therapy [6,7]. One study reported that five years after breast cancer diagnosis, the major determinants for quality of life are body mass index, relapse, socioeconomic deprivation, and comorbidities [6]. Specifically, it was found that body mass index, relapse, and socioeconomic deprivation were determinants of quality of life in younger women; in older women, comorbidities and socioeconomic deprivation were predictors of quality of life [6].

One study examined the impact of socioeconomic and clinical characteristics on quality of life in 188 breast cancer survivors five years after diagnosis. Quality of life was assessed with the Short Form Health Survey (SF-12), EORTCQLQ-C30, and EORTCQLQ-BR23 questionnaires. It was found that only age at diagnosis and comorbidities were determinants of quality of life [8]. In a more recent study it was reported that sociodemographic factors (level of education, marital status), age, comorbidities (e.g., diabetes, high cholesterol), stage of cancer, surgery, radiotherapy or hormone therapy do not have a significant effect on quality of life. The study investigated 77 patients and used the SF-36 scale [9]. Furthermore, a study of 544 female breast cancer patients revealed that demographic and clinical factors such as age, education, employment, marital status, income, chemotherapy, duration since treatment, and recurrence status had no impact on quality of life. EORTC QLQ-C30 and EORTC QLQ-BR23 was used for evaluating QOL [10]. The results of the above studies are conflicting, so there is a need for further research in order to investigate if demographic characteristics such as age, marital status, educational status, and occupational status influence quality of life in Greek breast cancer patients six months after the end of chemotherapy.

In addition, there are some studies that examine the impact of type of treatment on quality of life of breast cancer survivors [7,10,11]. It was reported that the type of breast cancer surgery did not influence any aspect of quality of life, only body image [10]. Chemotherapy has profound effects on quality of life (Yeo et al., 2018) regardless of whether it is administered in low or high doses [12]. Menopausal symptoms, whether they are due to natural menopause or due to cancer treatment, also affect quality of life [11]. Premenopausal breast cancer survivors have endocrine changes that affect quality of life. Women who become menopausal due to treatment have more severe menopausal symptoms compared to those who have a natural transition [7]. It was found that hormone therapy had no significant impacts on quality of life [9], but on the other hand causes some physical symptoms in breast cancer patients [13].

Anemia is a common adverse effect in cancer patients undergoing chemotherapy [14]. According to Schrijvers et al. (2010), anemia is defined as “a reduction of the hemoglobin (Hb) concentration, red blood cell count or packed cell volume below normal levels. Mild anemia is defined as an Hb ≤ 11 g/dL and Hb ≥ 10 g/dL, moderate anemia as Hb of ≤9.9 and ≥8 g/dL and severe anemia as an Hb of <8 g/dL” [15]. It is well documented that anemia induced by chemotherapy is associated with impaired quality of life in cancer patients [16]. Moreover, chemotherapy-induced neutropenia is a frequent complication in cancer patients because of myelosuppression [17]. This is associated with important effects on quality of life, because of hospitalization for febrile neutropenia and fear of hospitalization [18] Nowadays, myelotoxic regimes remain the current standard in care of breast cancer patients [17]. There is evidence that anemia and neutropenia are toxicities in breast cancer patients undergoing chemotherapy and most commonly appear 7–14 days after the completion of chemotherapy [19]. There is no evidence for a relationship between hemoglobin and neutrophil levels and quality of life in breast cancer survivors after the completion of chemotherapy. 

The aforementioned studies primarily investigated factors or determinants influencing quality of life and used several questionnaires for assessing quality of life, such as SF-36, EORTC-QLQ-C30, EORTC-QLQ-BR23, and FACT-B.

Undoubtedly, there is a growing interest in quality of life of breast cancer survivors after the acute phase of treatment [7,20]. In Greece, to the best of our knowledge, there are a limited number of studies assessing quality of life beyond chemotherapy in breast cancer patients (e.g., [21]), which is what stimulated this researcher’s interest in further investigation of this phenomenon.

The purpose of the present study was to assess breast cancer patients’ quality of life six months after the completion of adjuvant chemotherapy, and to investigate factors affecting this. Specifically, we aimed to assess the following research questions:Does age affect the domains of quality of life in breast cancer patients?Do different sociodemographic factors (educational status, occupational status, marital status) affect quality of life in breast cancer patients?Do treatment type, white blood cell count, hematocrit, hemoglobin, or menopausal status influence the domains of quality of life in breast cancer patients?

## 2. Material and Methods 

### 2.1. Design and Sample

A non-experimental descriptive study was conducted in one large hospital in a major Greek city. A convenience sample of 61 breast cancer outpatients was recruited. The inclusion criteria for patients were: A minimum age of 18, the ability to speak and read the Greek language, a histological breast cancer diagnosis, and chemotherapy treatment completed at most six months ago as an outpatient. Only patients who were not yet undergoing radiotherapy were eligible. Their performance status was assessed with Eastern Cooperative Oncology Group (ECOG) which is 0.

### 2.2. Data Collection

The study was approved by the general assembly of the Alexander Technological Educational Institute of Thessaloniki, which acts as an ethics committee. All potential subjects were approached by the researcher. She explained the purpose of the study to them and asked them to participate on a voluntary basis. She confirmed that their personal data would be kept anonymous. Informed consent was obtained from those who agreed to participate and the researcher interviewed all participating patients. Initially, 70 patients were approached; six refused to participate in the study and three questionnaires were excluded from further analysis because they had not been completed adequately. Thus, the response rate was 87.1%. Data were collected between March 2015 and March 2016.

### 2.3. Instruments

A questionnaire including an instrument for assessing quality of life and demographic and clinical information was used in order to collect data. Quality of life was assessed with the SF-36, which includes eight subscales. These subscales are the following: physical functioning, physical role, bodily pain, social functioning, mental health, emotional role, vitality, and general health perceptions. Each subscale includes 2–10 questions and has a score from 0–100. Zero is the lowest value and 100 is the highest. If the score is low, this points to an impaired quality of life or impaired quality in the domain described by the subscale (e.g., physical role, mental health, etc.). This instrument has been translated into numerous languages and has been tested for reliability and validity, showing satisfactory results [22]. In the present study, Cronbach’s alpha is 0.86 for the total scale.

### 2.4. Data Analysis

Statistical analysis was carried out using SPSS for Windows (IBM 21.00). To enable statistical analysis of the clinical data, participants were assigned post-hoc to different groups according to their levels of hemoglobin (normal range: 12–16 g/dL), hematocrit (normal range: 37–47%) and white cell count (normal count: 4500–11,000 μL of blood) [23]. This was done to assess whether quality of life was different between participants with hemoglobin, hematocrit and white cell counts within the normal range and participants with count in the abnormal range.

Descriptive statistics were used in order to analyze the demographic and clinical characteristics of the sample. Associations between the different demographic and clinical factors and their influence on SF-36 were tested, and were assessed using Pearson’s Chi-square or Fisher’s exact test where appropriate. The normality of the data was assessed using the Kolmogorov–Smirnov test, whereas the homogeneity of variances was tested using the Levene’s test. For identifying any outliers we used the outlier labeling rule with 2.2 as a multiplier [24] in every demographic and clinical variable. There was only one outlier identified (secondary education in physical role) and removed from analysis only for the physical role of the relevant group. Inferential statistics were used to assess the influence of the demographic and clinical variables on the SF-36 subscales. For the data that met the assumptions of the parametric tests, differences in the means were assessed using one-way ANOVA or *t* tests where appropriate. After significant ANOVAs, post-hoc comparisons with a Bonferonni correction were performed. Variables that deviated from normality were transformed using Box-Cox transformations [25]. For the variables that were successfully transformed into normality, parametric tests were implemented. For the variables that could not be transformed to normality, data were transformed back to raw and differences across the demographic or clinical groups were tested using the Kruskal–Wallis H test or Mann–Whitney U test where appropriate. For the significant H effects, post-hoc multiple comparisons with a Dunn–Bonferonni correction were performed. Two-way ANOVAs were used to analyze the interaction between two clinical variables on SF-36 (surgery type vs type of treatment). Where the assumptions were violated, bootstrapping using 1000 simulated samples was used for the estimation of marginal means and pairwise comparisons and the generated confidence intervals are reported. The reliability of the scales’ internal consistency reliability was estimated based on Cronbach’s alpha. The significance level was set to *p* < 0.05 and adjusted to multiple comparisons where appropriate.

## 3. Results

### 3.1. Demographic Characteristics and SF-36 Descriptives

The mean age of the patients was 51.52 ± 12.10. The majority of the patients were married (*n* = 47, 77%) and were high school graduates (*n* = 20, 32.8%). More than half of the sample had stage II cancer (*n* = 53, 86.9%) and underwent lumpectomy (*n* = 47, 77%). The vast majority of the sample had stage II cancer (*n* = 53, 86.9%) and almost half of the participants (*n* = 29, 47.5%) were menopausal. The demographic and clinical characteristics of the sample are presented in Table 1 and Table 2.

The associations between the various demographic and clinical characteristics were assessed. The associations between age and treatment type, surgery, hematocrit, hemoglobin, white count levels, and educational level were all non-significant (Appendix A, *p* > 0.05). The associations between age and menopausal status and age and work status were significant; older women tended to be menopausal and retired (*p* < 0.05). The associations between treatment type and the other demographic and clinical characteristics were also non-significant (Appendix A, *p* > 0.05). Finally, the association between type of surgery and most demographic and clinical variables were not significant (Appendix A, *p* > 0.05). However, there was a significant association between type of surgery and menopausal status (*p* = 0.04).

Table 3 shows the mean and standard deviations for the SF-36 subscales. The variables with lower means are the following: physical functioning (42.62 ± 39.09), mental health (53.57 ± 21.09), emotional role (56.74 ± 35.77), and vitality (56.15 ± 24.26).

### 3.2. Effect of Age Categories on SF-36 Subscales

Table 4 shows the effects of age (categories) on the different quality of life domains. The effect of age on physical role was significant [(F (3) = 3.26, *p* = 0.03]. Post-hoc comparisons revealed no significant differences across the age groups. However, there was a marginal difference between the 29–39 group and the 60+ group (*p* = 0.053), showing that the younger participants possibly had a better quality of life regarding physical role than participants that were over 60 years old. There were no other differences across the age groups for the other SF-36 subscales (*p* > 0.1) for quality of life domains. 

### 3.3. Effect of Menopause Status

Table 5 shows the effects of menopause status on the SF-36 scores. Τhe effect of menopause status on physical role score was significant [F (2) = 3.79, *p* = 0.03]. Post-hoc comparisons revealed a significant difference between the post-menopausal group and the group who were menopausal due to treatment (*p* = 0.045); the latter group reported a higher score than the former. Menopause status also had a marginal effect on emotional role [H (2) = 5.60, *p* = 0.06]; women who were menopausal due to treatment had a higher mean score than the other two groups, though not significantly.

### 3.4. The Effect of Education on the SF-36 Subscales

Table 6 shows the effects of education on the SF-36. There were no significant effects of education on quality of life. However, there were marginal effects on physical role [F (2) = 2.69, *p* = 0.08] and general health subscales [H (2) = 5.82, *p* = 0.054]. In both variables, patients with secondary education had higher mean scores than the other groups, followed by patients with higher education. However, these differences did not reach significance.

### 3.5. Effects of Work Status on the SF-36

Table 7 shows the main effects of work status on the SF-36. However, there were no significant effects on any of the scales (all *p* > 0.1).

### 3.6. Effects of Treatment Type (Whether Hormone Therapy Was Part of the Schema) on SF-36

Table 8 shows the effect of treatment type (with or without hormone therapy) on the SF-36 subscales. Patients that received hormone therapy in addition to surgery and chemotherapy reported a significantly higher quality of life in the bodily pain [U = 595.5, *p* = 0.04] and vitality [Welch’s F (1, 59) = 6.39, *p* = 0.01] subscales than the patients who underwent only surgery and chemotherapy. Patients who had hormone therapy also reported a marginally higher score in social functioning [U = 585.5, *p* = 0.08], however the difference was not significant. The two groups did not differ significantly on the other subscales (all *p* > 0.1).

### 3.7. Effects of Surgery Type on SF-36 Subscales

In Table 9, the differences of surgery type on SF-36 subscales are presented. There were no significant differences in any of the subscales. However, patients that underwent lumpectomy reported marginally higher scores in the general health subscale [t (59) = 1.78, *p* = 0.08]. In several subscales (physical role, bodily pain, social functioning, mental health), participants who had lumpectomy had higher scores than those who had mastectomy. However, these differences did not reach significance.

### 3.8. Effects of Hematocrit Levels on SF-36

Table 10 shows the differences between low levels and normal levels of hematocrit on the SF-36. There were no significant differences. However, participants with normal levels of hematocrit reported marginally higher scores in emotional role (U = 541.4, *p* = 0.05) and social functioning (U = 535, *p* = 0.07) than patients with low levels. In all subscales, patients with higher levels had higher scores in quality of life than the other group. However, these differences were not significant. 

### 3.9. Effects of Hemoglobin Levels on SF-36

Table 11 shows the mean and median SF-36 scores in low and normal levels of hemoglobin. There were no differences between the two groups.

### 3.10. Differences of White Cell Count Levels on SF-36

Table 12 shows the mean and median SF-36 scores in low and normal levels of white cell count. The were no significant differences between the low and normal level groups.

### 3.11. Interactions Between Type of Surgery and Treatment Type on SF-36 Subscales 

Given that there was a marginal association between the surgery type and type of treatment, the interaction between these variables and their effect independent of the other variable on two SF-36 subscales (bodily pain and vitality, on which type of treatment had a significant effect) were investigated. Regarding bodily pain, pairwise comparisons using bootstrapping on type of treatment showed that participants that received hormone therapy in addition to surgery and chemotherapy (mean: 63.85, standard error: 9.82, 95% CI: 44.39–83.76) had a significantly higher score than participants who did not receive this type of treatment (mean: 87.55, standard error: 3.03, 95% CI: 81.86–93.10, *p* = 0.03). Pairwise comparisons using bootstrapping on surgery showed that participants who had lumpectomy (mean: 82.89, standard error: 3.24, 95% CI: 76.40–88.84) did not significantly differ from participants who had mastectomy (mean: 68.5, standard error: 9.73, 95% CI: 50.42–88.75, *p* = 0.13). The main effects of type of treatment, surgery, and their interaction cannot be reported due data skewing.

Regarding vitality, the main effect of type of treatment was significant [F (1, 57) = 3.45, *p* = 0.004], whereas the main effect of surgery type was not significant [F (1, 57) = 0, *p* = 0.99] showing that type of surgery (lumpectomy or mastectomy) did not affect vitality. As is shown in Figure 1b, patients who had mastectomy that received hormone therapy had a higher mean score (mean: 73, SD: 18.14) in the vitality subscale than patients who had mastectomy that did not receive this type of treatment (mean: 36.25, SD: 24.62). However, the interaction between these variables was marginally significant [F (1, 57) = 3.45, *p* = 0.07]. 

## 4. Discussion

This study assessed the quality of life in Greek breast cancer patients and the factors (clinical, demographics) influencing this, six months after they completed their chemotherapy. It contributes to the growing body of evidence regarding quality of life in breast cancer survivors and provides important information for Greek oncology nurses, because describing this phenomenon is a significant step toward appropriate interventions. In this study we found that age, menopausal status, and hormone therapy influenced certain domains of quality of life in breast cancer patients.

Breast cancer patients in this study sample experienced low levels of physical functioning, mental health, emotional role, and vitality. This is consistent with the results of other studies [21,26].

We examined the associations between demographic and clinical characteristics and we found that there is statistically significant difference between age and menopausal status, as well as between age and work status; the older women were post-menopausal and retired. This is an expected outcome because older women tended to be retired. Also, we found that there was an association between type of surgery and menopausal status. To the best of our knowledge, the type of hormonal therapy after the surgery was associated with menopausal status but there is not clear evidence that the type of breast surgery affected menopausal status [27]. This is influenced by the kind of therapy after the surgery (chemotherapy, radiation therapy, hormone therapy). Hence, there is a need for further research in order to draw a safe conclusion. 

The influence of various demographic characteristics in breast cancer survivors’ quality of life was studied previously and was also investigated in the present study. We found that age affected one subscale of quality of life, physical role. This is inconsistent with the findings of other studies [9,10], and is partially consistent with one study which reported that only age at diagnosis was a determinant of the quality of life [8]. This discrepancy across the literature might be explained by the differences in the sample size or different questionnaires that were used across the studies. We also find a marginal effect of age on physical role; which younger participants might have had a better quality of life in physical role than the older women (>60 years old). Although this is an expected outcome, in previous studies it was considered that early age (<50 years) was a predictive factor of a worse quality of life [9,28]. There is a need for further research in order to clarify the influence of age in quality of life of breast cancer survivors, especially in Greek patients.

We found that educational status has no effect on quality of life. This finding is consistent with previous studies [9,10]. A marginal effect of educational status on the physical role and general health subscales was found. A possible explanation for these results is the small sample size in educational status groups. Also, we found that work did not influence any domain of quality of life. This is in line with a previous study which reported that employment had no impact on quality of life [10].

It is worthwhile to note that we did not find any influence of demographic on the mental health domain of quality of life. This is in accordance with the findings of other studies [9,20]. There is a need for further research in Greece to clarify if the aforementioned factors influence mental health; more specific questionnaires should be used. 

The present study also examined whether clinical characteristics influence quality of life and its domains. Menopausal status has a significant effect on physical role. This association has also been reported in China and in Australia in breast cancer survivors [11,29]. Also, we found women experiencing menopausal symptoms due to treatment had a higher quality of life than post-menopausal women. A possible explanation for this finding is the fact that women in this sample were younger and they experienced menopausal symptoms after initial treatment [30] so, the menopausal status per se may not affect their quality of life in a great manner. As is reported above, we found an association between age and menopausal status and this finding strengthens our explanation. Moreover, we found another marginal but not significant effect of menopausal status on emotional role, in which women who were menopausal due to treatment had better emotional role scores than the others. This might be explained their young age. In a previous systematic review, it was considered that younger age was associated with depression [28]. Future studies could test whether symptoms of depression mediate the association between age and quality of life in Greek breast cancer patients. Furthermore, we found an effect of treatment type on the quality of life subscales. Specifically, we found that hormone therapy on top of surgery and chemotherapy seemed to be associated with better quality of life regarding bodily pain and vitality. This finding is inconsistent with the findings of another study [9]. Seventy breast cancer survivors were examined, but they did not refer many details of duration of survival after their initial treatment. Also, they used the same scale as the present study but they did not refer specifically to influences of hormone therapy and other types of therapy on quality of life subscales. Also, this finding is partially consistent with the findings of Mortada et al. [13] who reported that the hormonal therapy group had higher scores in all domains of quality of life (they used EORTC-QLQ30 and QLQ-BC23). This conflicting finding might be explained by the fact that the studies used different scales and had different sample sizes. Moreover, in our study there was a shorter period between the initial type of treatment (surgery, chemotherapy, and hormonal therapy) and the interview, so the participants would not have noticed as many disruptions in all domains of their quality of life. It is noteworthy that the marginal effect of hormone therapy on social function is explained by the previous finding. We can hypothesize that these patients had no bodily pain and had better vitality so it was easier for them to have a better social life.

No significant association was found between surgery type and quality of life subscales. Similarly, Dadjoul et al. [9] did not find any influence of type of surgery on overall quality of life. Another study in long-term breast cancer survivors (5 years) reported that surgical modality has no significant impact on any quality of life domains except body image [10]. Once again, it is worth mentioning the marginally higher general health scores in patients with lumpectomy and the non-significant higher scores in some subscales (physical role, bodily pain, social functioning, and mental health). This result might be explained by the small sample size (in the present study a few patients were undergoing mastectomy). Further study is needed to make clear the effect of type surgery in breast cancer patients in Greece.

Regarding the influence of hematocrit and hemoglobin in quality of life, we showed that there is not a statistically significant difference. This is an expected outcome, because our patients did not experience severe anemia so it did not affect their quality of life. On the other hand, they had finished their chemotherapy less than six months before the study, and as we know anemia is a serious adverse effect during chemotherapy [16]. To the best of our knowledge this is the first study that examines the association between anemia and quality of life in breast cancer survivors. Therefore, it is not possible for us to compare these results. Further study is needed with larger sample sizes in order to ascertain if there is association between hematocrit and hemoglobin levels on quality of life in breast cancer survivors in Greece and elsewhere.

We also found a marginally higher effect of hematocrit in emotional role and social functioning where participants with low hematocrit levels had lower scores than those with normal levels. This is explained by the fact that low levels of hematocrit influence patients’ daily activities and impact their quality of life [16]. Future studies could test this hypothesis by examining the influence of low levels of hematocrit on different domains of quality of life.

Our study showed that there was no impact of white cell count levels on domains of quality of life. This was also an expected outcome because our sample had no cases of severe neutropenia. There is a need for further research to clarify if white cell count levels influence quality of life.

Analysis yielded an interesting result. Breast cancer survivors who received hormone therapy in addition to surgery and chemotherapy had higher scores in the bodily pain domain of quality of life than others who did not receive this type of therapy. This is not consistent with the finding of one study that examined 186 women who took hormonal therapy after surgery and found that women taking hormonal therapy had more severe joint or bone pain than women not taking hormonal therapy [31]. This discrepancy might be attributable to the different sample size, different culture, and different methodologies used by the two studies. Undoubtedly, there is a need for further research in order to discriminate the influences of hormonal therapy and previous therapy on breast cancer survivors’ quality of life.

Another interesting result is the interaction of hormone therapy and other treatment types. Women who had received hormone therapy in addition to surgery and chemotherapy had better quality of life regarding bodily pain. Although hormone therapy causes joint pain, this result is inconsistent with the known results [31]. This discrepancy might be explained by the small size of the present study.

Finally, we found that the effect of type of treatment on vitality was significant. It should be pointed out that vitality is linked to fatigue. Fatigue is a common and disabling symptom in cancer patients and in patients’ survivorship. It can be measured by the vitality subscale of SF-36 questionnaire [32]. Therefore the above result is an expected outcome because fatigue is a symptom associated with chemotherapy, radiation therapy, surgery, and hormonal therapy and often persists after the completion of therapy [31,33].

### Limitations of the Study

This study has some limitations. It was conducted in one hospital located in a major Greek city and the sample was relatively small, so the results cannot be generalized with respect to the entire Greek population. Another limitation was the cross-sectional nature of the present study; we could not assess the trajectory of quality of life at the end the chemotherapy, six months on, one year on, or five years on. A future study in Greece employing a longitudinal design could provide more clear conclusions. Although we studied the influence of demographic and clinical characteristics, the present study did not investigate some other important clinical characteristics such as type of chemotherapy regimen, its duration, etc. However, the results provide valuable information for the issue at hand, and illustrate the great need for further longitudinal studies in order to draw reliable conclusions. Despite these limitations our study has one significant strength: To our knowledge, this is the first population-based study to investigate factors influencing quality of life in breast cancer survivors in Greece, where culture and lifestyle are significantly different from those in western populations.

## 5. Conclusions

The results of the present study have shown, once again, that the quality of life of breast cancer patients is affected. The majority of the survivors have low scores for physical functioning, mental health, emotional role, and vitality. Some demographic and clinical characteristics influenced some domains of breast cancer survivors’ quality of life; age, menopausal status, and type of treatment are factors which influenced quality of life in this study group. The domains influenced were physical role, bodily pain and vitality. Health care professionals need to be more aware of quality of life issues within this group of cancer patients in order to meet their needs, both following acute treatment and also for the duration of their struggle with this disease. They should arrange care plans and take into account these factors in order to find ways beside pharmacological interventions to increase these patients’ quality of life. 

Further research examining the trajectory of quality of life in breast cancer patients in other oncology hospitals in Greece could add important information to the Greek oncology nursing literature.

## Figures and Tables

**Figure 1 diseases-07-00026-f001:**
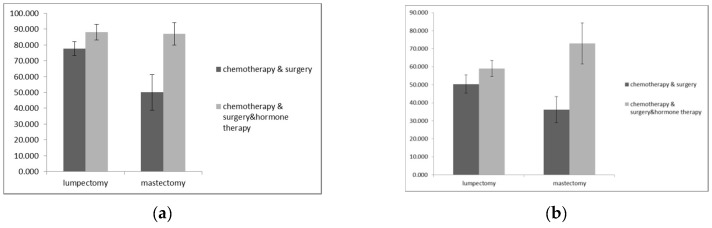
(**a**) Mean bodily pain scores according to treatment type across the surgery groups. Error bars show the standard error of the mean; (**b**) mean vitality scores according to treatment type across the surgery groups. Error bars show the standard error of the mean.

**Table 1 diseases-07-00026-t001:** Demographic characteristics of the sample.

Characteristics	N	%
**Age**		
29–39	12	19.7
40–49	20	32.8
49–59	13	21.3
>60	16	26.2
**Marital Status**		
Single	5	8.2
Married	47	77
Divorced	2	3.3
Widowed	7	11.5
**Educational Status**		
Primary school	15	25
Secondary education	23	38.3
Higher education	22	36.7
**Occupational Status**		
Blue collar	7	11.5
White collar	17	27.9
Homemaker	26	42.6
Retired	11	18

**Table 2 diseases-07-00026-t002:** Clinical characteristics of the sample.

Characteristics	N	%
**Cancer Stage**		
Stage I	5	8.2
Stage II	53	86.9
Stage III	3	4.9
**Menopausal Status**		
Pre-menopausal	14	23
Post-menopausal	29	47.5
Menopausal due to treatment	18	29.5
**Surgery**		
Lumpectomy	47	77
Mastectomy	14	23
**Treatment Type**		
Chemotherapy and surgery	30	49.2
Chemotherapy, surgery, and hormone therapy	31	50.8
**Other**	**Mean**	
Hematocrit	45 ± 38.04	
Hemospherin	14.60 ± 12.39	
White cell count	14,700 ± 5363	

**Table 3 diseases-07-00026-t003:** Mean ± SD, median, and IQR of SF-36 scores.

Variables	Mean ± SD	Median (IQR)
Physical functioning	42.62 ± 39.08	25 (75)
Physical role	68.60 ± 20.27	70 (25)
Bodily pain	80.93 ± 24.06	100 (30)
General health	64.42 ± 22.51	65 (40)
vitality	56.14 ± 24.26	60 (35)
Social functioning	65.68 ± 31.75	75 (63)
Emotional role	56.73 ± 35.76	33 (67)
Mental health	53.57 ± 21.09	52 (32)

**Table 4 diseases-07-00026-t004:** Significant effects of age on SF-36 subscales.

SF-36	29–39	40–49	50–59	60+	H or F (df) ^4^	*p*-Value
Physical functioning	43.75 ± 44.11 ^1^	36.25 ± 40.13 ^1^	51.92 ± 37.45 ^1^	42.19 ± 37.33 ^1^	H (3) = 1.58	0.66
	25 (100) ^2^	25 (75) ^2^	50 (62.5) ^2^	25 (68.75) ^2^		
Physical role	75.83 ± 20.65 ^1^	71.75 ± 18.59 ^1^	72.69 ± 13.94 ^1^	55.94 ± 13.94 ^1^	F (3, 57) = 3.26	0.03
	80 (23.75) ^2^	75 (20) ^2^	70 (20) ^2^	60 (33.75) ^2^		
Bodily pain	86.67 ± 20.15 ^1^	82 ± 18.24 ^1^	83.08 ± 22.13 ^1^	73.75 ± 33.44 ^1^	H (3) = 1.00	0.80
	100 (27.5) ^2^	80 (30) ^2^	100 (30) ^2^	95 (62.5) ^2^		
General health	64.58 ± 26.15 ^1^	63 ± 24.57 ^1^	70.77 ± 20.60 ^1^	60.94 ± 19.25 ^1^	F (3, 57) = 0.49	0.69
	70 (47.5) ^2^	67.5 (47.5) ^2^	75 (37.5) ^2^	62.5 (33.75) ^2^		
Vitality ^3^	62.5 ± 26.42 ^1^	55.5 ± 26.75 ^1^	57.69 ± 19.43 ^1^	50.94 ± 23.90 ^1^	F (3, 57) = 0.59	0.62
	67.5 (47.5) ^2^	62.5 (45) ^2^	60 (25) ^2^	50 (28.75) ^2^		
Social functioning	72.83 ± 28.73 ^1^	58.65 ± 30.18 ^1^	75.92 ± 28.18 ^1^	60.81 ± 37.62 ^1^	H (3) = 2.97	0.40
	75 (59.75) ^2^	50 (56.75) ^2^	75 (44) ^2^	68.5 (63) ^2^		
Emotional role	58.25 ± 45.27 ^1^	53.20 ± 34.99 ^1^	58.92 ± 33.88 ^1^	58.25 ± 33.47 ^1^	H (3) = 0.30	0.96
	66.5 (91.75) ^2^	33 (67) ^2^	67 (67) ^2^	50 (67) ^2^		
Mental health	56.33 ± 22.07 ^1^	52.80 ± 21.51 ^1^	53.54 ± 19.22 ^1^	52.5 ± 23.09 ^1^	F (3, 57) = 0.09	0.97
	54 (40) ^2^	56 (24) ^2^	60 (32) ^2^	48 (37) ^2^		

^1^ Mean ± standard deviation; ^2^ median (interquartile range); ^3^ analysis was implemented on transformed data (mean, median, and measures of dispersion on raw data are presented to enable comparisons across subscales); ^4^ H: Kruskal–Wallis, F: one-way ANOVA.

**Table 5 diseases-07-00026-t005:** Significant effects of menopause status on SF-36 subscales.

SF-36	Pre-Menopausal	Post-Menopausal	Menopausal due to Treatment	H or F (df) ^4^	*p*-Value
Physical functioning	25 ± 36.69 ^1^	43.97 ± 35.77 ^1^	54.17 ± 43.09 ^1^	H (2) = 4.90	0.09
	0 (50) ^2^	25 (35.77) ^2^	50 (81.25) ^2^		
Physical role ^3^	73.93 ± 19.33 ^1^	62.07 ± 18.58 ^1^	75 ± 21.35 ^1^	F (2, 58) = 3.79	0.03
	77.50 (17.50) ^2^	65 (17.50) ^2, b^	80 (28.75) ^2, a^		
Bodily pain	81.43 ± 20.71 ^1^	76.90 ± 28.55 ^1^	87.22 ± 17.42 ^1^	H (2) = 1.31	0.52
	85 (32.50) ^2^	90 (35) ^2^	100 (30) ^2^		
General health ^3^	60 ± 25.72 ^1^	64.48 ± 21.52 ^1^	67.78 ± 22.18 ^1^	F (2, 58) = 0.39	0.68
	62.5 (47.5) ^2^	60 (32.5) ^2^	72.5 (46.25) ^2^		
Vitality	51.43 ± 24.21 ^1^	54.83 ± 21.40 ^1^	61.95 ± 28.60 ^1^	F (2, 58) = 0.82	0.45
	47.5 (45) ^2^	60 (27.50) ^2^	70 (46.25) ^2^		
Social functioning	57.93 ± 33.54 ^1^	66.28 ± 34.45 ^1^	70.78 ± 25.80 ^1^	H (2) = 1.24	0.54
	50 (66) ^2^	75 (63) ^2^	75 (50) ^2^		
Emotional role	42.71 ± 40.18 ^1^	52.97 ± 30.25 ^1^	72.11 ± 36.73 ^1^	H (2) = 5.60	0.06
	33 (100) ^2^	33 (34) ^2^	100 (67) ^2^		
Mental health	49.71 ± 23.83 ^1^	54.62 ± 21.55 ^1^	54.89 ± 18.85 ^1^	F (2, 58) = 0.29	0.75
	50 (26) ^2^	52 (34) ^2^	62 (29) ^2^		

^1^ Mean ± standard deviation; ^2^ median (interquartile range); ^3^ analysis was implemented on transformed data (mean, median, and measures of dispersion on raw data are presented to enable comparisons across subscales); ^4^ H: Kruskal–Wallis, F: one-way ANOVA. ^a^ Significantly different from the post-menopausal group; ^b^ significantly different from the menopausal due to treatment group.

**Table 6 diseases-07-00026-t006:** Significant effects of education on SF-36 subscales.

SF-36	Primary School	Secondary Education	Higher Education	H or F (df) ^4^	*p*-Value
Physical functioning	31.67 ± 35.95 ^1^	55.44 ± 37.66 ^1^	34.09 ± 38.99 ^1^	H (2) = 4.56	0.1
	25 (50) ^2^	75 (75) ^2^	25 (62.50) ^2^		
Physical role ^5^	61.33 ± 14.82 ^1^	75.68 ± 14.82 ^1^	69.77 ± 23.32 ^1^	F (2, 56) = 2.69	0.08
	65 (15) ^2^	75 (15) ^2^	75 (31.25)		
Bodily pain	66.67 ± 30 ^1^	82.61 ± 24.16 ^1^	86.82 ± 17.83 ^1^	H (2) = 4.20	0.12
	70 (50) ^2^	100 (30) ^2^	100 (30) ^2^		
General health	55.67 ± 22.90 ^1^	72.39 ± 21.31 ^1^	61.36 ± 21.83 ^1^	H (2) = 5.82	0.05
	60 (30) ^2^	85 (35) ^2^	65 (40) ^2^		
vitality	52.67 ± 25.83 ^1^	61.52 ± 24.24 ^1^	53.18 ± 23.88 ^1^	F (2, 57) = 0.87	0.43
	50 (40) ^2^	65 (30) ^2^	55 (33.75) ^2^		
Social functioning	61.53 ± 38.27 ^1^	64.61 ± 32.39 ^1^	68.9 ± 27.02 ^1^	H (2) = 0.16	0.93
	50 (63) ^2^	70 (63) ^2^	75 (52.25) ^2^		
Emotional role	59.93 ± 28.91 ^1^	56.44 ± 36.94 ^1^	52.91 ± 39.46 ^1^	H (2) = 0.43	0.81
	67 (67) ^2^	33 (67) ^2^	33 (67) ^2^		
Mental health	52.27 ± 16.46 ^1^	54.78 ± 26.65 ^1^	53.64 ± 18.42 ^1^	F (2, 57) = 0.06	0.94
	48 (28) ^2^	64 (44) ^2^	58 (28) ^2^		

^1^ Mean ± standard deviation; ^2^ median (interquartile range); ^3^ analysis was implemented on transformed data (mean, median, and measures of dispersion on raw data are presented to enable comparisons across subscales); ^4^ H: Kruskal–Wallis, F: one-way ANOVA; ^5^ an outlier of secondary education in physical role was removed from the analysis.

**Table 7 diseases-07-00026-t007:** Effects of work status on SF-36 subscales.

SF-36	Blue Collar	White Collar	Housemaker	Retired	H or F (df) ^4^	*p*-Value
Physical functioning	35.71 ± 47.56 ^1^	38.24 ± 47.56 ^1^	43.27 ± 37.79 ^1^	52.27 ± 39.46 ^1^	H (3) = 1.40	0.71
	0 (100) ^2^	25 (87.5) ^2^	25 (75) ^2^	50 (75) ^2^		
Physical role ^3^	72.14 ± 20.59 ^1^	73.82 ± 21.47 ^1^	66.54 ± 19.48 ^1^	63.18 ± 20.65 ^1^	F (3, 57) = 0.98	0.41
	75 (35) ^2^	75 (17.5) ^2^	70 (21.25) ^2^	60 (35) ^2^		
Bodily pain	80 ± 20^1^	86.47 ± 18 ^1^	77.31 ± 27.06 ^1^	81.82 ± 28.22 ^1^	H (3) = 1.03	0.79
	70 (30) ^2^	100 (25) ^2^	90 (40) ^2^	100 (30) ^2^		
General health	66.43 ± 22.12 ^1^	60.88 ± 19.55 ^1^	63.85 ± 25.51 ^1^	70 ± 21.21 ^1^	F (3, 57) = 0.38	0.77
	60 (50) ^2^	65 (32.5) ^2^	67.5 (42.5) ^2^	80 (35) ^2^		
Vitality	53.57 ± 27.65 ^1^	56.47 ± 21.71 ^1^	57.5 ± 27.18 ^1^	54.09 ± 21.43 ^1^	F (3, 57) = 0.08	0.97
	55 (55) ^2^	60 (32.5) ^2^	62.5 (36.25) ^2^	65 (25) ^2^		
Social functioning	62.43 ± 29.83 ^1^	66.82 ± 28.03 ^1^	71.08 ± 33 ^1^	53.38 ± 35.80 ^1^	H (3) = 2.54	0.47
	50 (63) ^2^	75 (63) ^2^	75 (53.35) ^2^	50 (62) ^2^		
Emotional role	66.57 ± 33.5 ^1^	54.77 ± 40.80 ^1^	57.62 ± 67 ^1^	51.56 ± 31.29 ^1^	H (3) = 0.84	0.84
	67 (67) ^2^	33 (67) ^2^	50 (67) ^2^	33 (34) ^2^		
Mental health	56 ± 17.74 ^1^	54.12 ± 19.49 ^1^	53.08 ± 25.21 ^1^	52.36 ± 16.82 ^1^	F (3, 57) = 0.05	0.99
	48 (32) ^2^	60 (30) ^2^	56 (35) ^2^	48 (20) ^2^		

^1^ Mean ± standard deviation; ^2^ median (interquartile range); ^3^ analysis was implemented on transformed data (mean, median, and measures of dispersion on raw data are presented to enable comparisons across subscales); ^4^ H: Kruskal–Wallis, F: one-way ANOVA.

**Table 8 diseases-07-00026-t008:** Effects of treatment type on SF-36 subscales.

SF-36	Chemotherapy and Surgery	Chemotherapy, Surgery, and Hormone Therapy	U or t (df) ^4^	*p*-Value
Physical functioning	36.57 ± 42.41 ^1^	48.39 ± 35.32 ^1^	U = 575.5	0.1
	25 (81.25) ^2^	25 (50) ^2^		
Physical role ^3^	65.33 ± 23.15 ^1^	71.77 ± 16.81 ^1^	t (59) = 1	0.32
	70 (30) ^2^	75 (25) ^2^		
Bodily pain	74 ± 27.74 ^1^	87.74 ± 17.83 ^1^	U = 595.5	0.04
	70 (40) ^2^	100 (30)		
General health ^3^	65.82 ± 22.40 ^1^	63.07 ± 22.9 ^1^	t (59) = 0.47	0.64
	70 (37.5) ^2^	65 (40) ^2^		
Vitality	48.5 ± 26.23 ^1^	63.55 ± 19.93 ^1^	F (1, 59) = 6.39 ^5^	0.01
	42.5 (40) ^2^	65 (25) ^2^		
Social functioning	57.83 ± 33.91 ^1^	73.29 ± 27.99 ^1^	U = 585.5	0.08
	50 (66) ^2^	75 (50) ^2^		
Emotional role	58.80 ± 37.93 ^1^	54.74 ± 34.06 ^1^	U = 437.7	0.67
	50 (67) ^2^	33 (67) ^2^		
Mental health	51.20 ± 21.87 ^1^	55.87 ± 20.41 ^1^	t (59) = −0.86	0.39
	48 (37) ^2^	60 (28) ^2^		

^1^ Mean ± standard deviation; ^2^ median (interquartile range); ^3^ analysis was implemented on transformed data (mean, median, and measures of dispersion on raw data are presented to enable comparisons across subscales); ^4^ U: Mann–Whitney, t: independent samples *t* test; ^5^ Welch’s F-test.

**Table 9 diseases-07-00026-t009:** Effects of surgery type on SF-36 subscales.

SF-36	Lumpectomy	Mastectomy	U or t (df) ^4^	*p*-Value
Physical functioning	42.02 ± 40.08 ^1^	44.64 ± 36.92 ^1^	U = 262	0.25
	25 (75) ^2^	25 (56.25) ^2^		
Physical role	70.11 ± 20.86 ^1^	63.57 ± 17.91 ^1^	t (59) = 1.06	0.29
	75 (25) ^2^	67.5 (32.5) ^2^		
Bodily pain	82.24 ± 23.15 ^1^	76.43 ± 27.35 ^1^	U = 286.5	0.43
	100 (30) ^2^	80 (30) ^2^		
General health ^3^	66.92 ± 22.85 ^1^	56.97 ± 19.82 ^1^	t (59) = 1.78	0.08
	70 (35) ^2^	52.5 (33.75) ^2^		
Vitality	54.26 ± 23.75 ^1^	62.5 ± 25.78 ^1^	t (59) = −1.12	0.27
	60 (30) ^2^	67.5 (50) ^2^		
Social functioning	66.92 ± 32.14 ^1^	61.57 ± 31.22 ^1^	U = 292	0.51
	75 (63) ^2^	75 (47.25) ^2^		
Emotional role	55.94 ± 37.59 ^1^	59.43 ± 29.93 ^1^	U = 348.5	0.72
	33 (67) ^2^	50 (67) ^2^		
Mental health	53.87 ± 21.10 ^1^	52.57 ± 21.85 ^1^	t (59) = 0.20	0.84
	56 (32) ^2^	46 (41) ^2^		

^1^ Mean ± standard deviation; ^2^ median (interquartile range); ^3^ analysis was implemented on transformed data (mean, median, and measures of dispersion on raw data are presented to enable comparisons across subscales); ^4^ U: Mann–Whitney, t: independent samples *t* test.

**Table 10 diseases-07-00026-t010:** Effects of hematocrit levels on SF-36 subscales.

SF-36	Low Levels (*n* = 21)	Normal Levels (*n* = 40)	U or t (df) ^4^	*p*-Value
Physical functioning	36.91 ± 42.29 ^1^	45.63 ± 37.51 ^1^	U = 490	0.27
	25 (87.5) ^2^	25 (50) ^2^		
Physical role ^3^	68.10 ± 21.71 ^1^	68.88 ± 19.76 ^1^	t (59) = −0.09	0.93
	75 (35) ^2^	70 (18.75) ^2^		
Bodily pain	80.95 ± 23 ^1^	81 ± 24.89 ^1^	U = 426	0.92
	100 (35) ^2^	95 (30) ^2^		
General health	63.81 ± 21.85 ^1^	64.75 ± 23.12 ^1^	U = 437.5	0.79
	70 (42.5) ^2^	65 (35) ^2^		
Vitality ^3^	55.24 ± 24.00 ^1^	56.63 ± 24.68 ^1^	t (59) = −0.25	0.80
	60 (25) ^2^	60 (25) ^2^		
Social functioning	56.43 ± 30.60 ^1^	70.55 ± 31.63 ^1^	U = 535.00	0.07
	50 (50.5) ^2^	75 (50) ^2^		
Emotional role	44.33 ± 33.98 ^1^	63.25 ± 33.97 ^1^	U = 541.5	0.05
	33 (34) ^2^	67 (67) ^2^		
Mental health	44.33 ± 33.97 ^1^	63.25 ± 35.26 ^1^	F (1, 55.01) = 0.77 ^5^	0.39
	33 (34) ^2^	67 (67) ^2^		

^1^ Mean ± standard deviation; ^2^ median (interquartile range); ^3^ analysis was implemented on transformed data (mean, median, and measures of dispersion on raw data are presented to enable comparisons across subscales); ^4^ U: Mann–Whitney, t: independent samples *t* test; ^5^ Welch’s F-test.

**Table 11 diseases-07-00026-t011:** Effects of hemoglobin levels on the SF-36 subscales.

SF-36	Low Levels (*n* = 19)	Normal Levels (*n* = 42)	U or t (df) ^4^	*p*-Value
Physical functioning	36.84 ± 40.28 ^1^	45.24 ± 38.75 ^1^	U = 453.5	0.38
	25 (75) ^2^	25 (81.25) ^2^		
Physical role ^3^	70.26 ± 20.45 ^1^	67.86 ± 20.40 ^1^	t (59) = 0.43	0.67
	75 (30) ^2‘^	70 (25) ^2^		
Bodily pain	83.16 ± 22.12 ^1^	80 ± 25.09 ^1^	U = 377.5	0.72
	100 (30) ^2^	95 (30) ^2^		
General health	67.37 ± 21.56 ^1^	63.10 ± 23.06 ^1^	t (59) = 0.68	0.50
	70 (40) ^2‘^	65 (37.5) ^2^		
Vitality	56.32 ± 22.04 ^1^	56.07 ± 25.46 ^1^	U = 417	0.78
	60 (25) ^2^	60 (35) ^2^		
Social functioning	57.79 ± 28.41 ^1^	69.26 ± 32.85 ^1^	U = 500	0.11
	50 (38) ^2^	75 (50) ^2^		
Emotional role	45.53 ± 31.95 ^1^	61.81 ± 36.60 ^1^	U = 497	0.11
	33 (34) ^2^	67 (67) ^2^		
Mental health	51.58 ± 18.28 ^1^	54.48 ± 22.40 ^1^	t (59) = −0.49	0.62
	48 (24) ^2^	56 (33) ^2^		

^1^ Mean ± standard deviation; ^2^ Median (interquartile range); ^3^ analysis was implemented on transformed data (mean, median, and measures of dispersion on raw data are presented to enable comparisons across subscales); ^4^ U: Mann–Whitney, t: independent samples *t* test.

**Table 12 diseases-07-00026-t012:** Effects of white cell count levels on SF-36 subscales.

SF-36	Low Levels (*n* = 23)	Normal Levels (*n* = 36)	U or t (df) ^4^	*p*-Value
Physical functioning	43.48 ± 40.04 ^1^	43.06 ± 39.92 ^1^	U = 415	0.99
	25 (75) ^2^	25 (93.75) ^2^		
Physical role ^3^	69.57 ± 20.00 ^1^	67.78 ± 21.23 ^1^	t (57) = 0.32	0.75
	70 (25) ^2^	72.5 (25) ^2^		
Bodily pain	80.44 ± 24.95 ^1^	81.94 ± 24.36 ^1^	U = 328.5	0.81
	100 (40) ^2^	100 (30) ^2^		
General health ^3^	60 ± 22.21 ^1^	65.97 ± 22.51 ^1^	t (57) = 0.76	0.32
	55 (35) ^2^	70 (33.75) ^2^		
Vitality	56.09 ± 25.80 ^1^	56.81 ± 23.40 ^1^	t (57) = −0.11	0.91
	60 (35) ^2^	60 (35) ^2^		
Social functioning	62.96 ± 34.09 ^1^	66.92 ± 30,84 ^1^	U = 438.5	0.70
	75 (63) ^2^	75 (59.75) ^2^		
Emotional role	54.96 ± 39.79 ^1^	58.25 ± 34.28 ^1^	U = 436.5	0.71
	33 (67) ^2^	50 (67) ^2^		
Mental health	50.44 ± 18.98 ^1^	56 ± 22.57 ^1^	t (57) = −0.98	0.33
	48 (28) ^2^	60 (31) ^2^		

^1^ Mean ± standard deviation; ^2^ median (interquartile range); ^3^ analysis was implemented on transformed data (mean, median, and measures of dispersion on raw data are presented to enable comparisons across subscales); ^4^ U: Mann–Whitney, t: independent samples *t* test.

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
