# Peer review of "Factors Influencing Quality of Life in Breast Cancer Patients Six Months after the Completion of Chemotherapy"

_diseases, 2019, doi:10.3390/diseases7010026_

Round 1

Reviewer 1 Report

Title: Factors influencing the quality of life in breast cancer patients six months after completion of

         chemotherapy

General Comment

The authors conducted a prospective study to assess the impact on QOL parameters in the immediate six months after completion of adjuvant therapy in subjects with early breast cancer. A number of previously published papers have addressed the same issue though, perhaps, none in Greece. As such, one must ensure that any “new” publication truly adds to the published literature. Overall, the study appears to have only modest relevance as the findings are not novel.

There are a number of issues with this paper which may affect the validity of the results. The following comments are provided with the intent of clarifying author assertions and/or improving the overall quality of the paper. Below are a number of issues identified sequentially as they appear in the paper rather than based on a perceived level of importance.

Specific comments

Page 1, Abstract.      The text of this section, specifically the terms “significant effect” and      “significantly higher” do not provide a clear understanding of what this      really means. There is also an apparent contradiction as in “…scored significantly      higher in body pain thus reporting lower pain …”. Consider      modifying the text so it is clearly written.

Page 2,      paragraph 4. Sentence 2, as written, infers that emotional distress,      fatigue, and premature menopause are the principal mediators or factors      associated with body/joint pain, etc. Is this really so?

Page 3

a.   1st sentence. Text indicates that SF-36 has 8 subscales; only 7 are listed. It is noted here that the rating scale ranges from 0 to100; the higher the rating relates to a lower or poorer QOL.

b.    Data analysis.

(1)   provide an explanation why post-hoc groups were categorized based also on hematocrit and WBC.

(2)   why was mean age used instead of median age?

(3)   reviewer perspective – the sample population appears to be disproportionately tilted toward relatively young (premenopausal) women. Is the age of most new diagnoses of breast cancer in Greece different from other European countries and North America which is predominantly postmenopausal?

Page 4

a.     Table 2, treatment type. Did any patient get post-surgical radiation?

b.    Table 3. SDs are very wide.

c.     Specifically how should the statement “Age had a significant effect on the role physical…” be interpreted?

d.    Table 4. According to the data presented, older patients (i.e., >60 years of age) had better role physical ratings than the other age groups. Was part of the explanation for this finding explained by extent of surgery (i.e., total mastectomy) thus foregoing radiation?

Page 5. Reviewer      perspective. The results cannot be considered accurate because standard of      care for patients undergoing breast-conserving surgery is the inclusion of      post-operative radiation to the remaining breast tissue (which could have      a significant effect of QOL in the short term). Was none given?

Page 6.

a.     Discussion. Frequently mentioned is the use of the terms “low score” which was initially defined in the Instrument section (i.e., lower score being associated with better QOL). In this section the use of these two words appears to be associated with poorer QOL. Please reconcile this discrepancy. Paragraph 3 is another example of this confusing message.

b.    Paragraph 8, 1st sentence. What exactly is being described here?

Summary

Although the results of the study show treatment-associated alterations of QOL parameters in patients with newly diagnosed breast cancer, the study itself is weak from design and assessment perspectives. Furthermore, the suggestion that healthcare professionals need to be more aware of QOL issues is a relatively callous statement. All things considered, this reviewer was less than satisfied with the publication-quality of the paper.

Author Response

REVIEWER   1

ANSWERS

Page 1, Abstract.  The text of this section, specifically   the terms “significant effect” and “significantly higher” do not provide a   clear understanding of what this really means. There is also an apparent   contradiction as in “…scored significantly  higher in   body pain thus reporting lower pain …”. Consider        modifying the text so it is clearly written.

Abstract is corrected in this point. Generally,   abstract is changed due to changes in the data analysis strategy and therefore   results. This point is changed as following “The mean age of the patients was 51.52±12.10. The effect of age on the Role Physical was   significant (p =   0.003). Τhe effect of menopausal status on “role physical”   was found to be significant (p = 0.003). Post-hoc comparisons revealed a significant   difference between post-menopausal and menopausal due to treatment group (p =   .045) in which the second group reported a higher score than the first. Regarding the treatment type, patients who   received  hormone therapy in addition   to surgery and chemotherapy reported a significant higher quality of life in   bodily pain (p=0.04) and vitality (p=0.04) than patients who underwent only   surgery and chemotherapy.”

Page 2, paragraph 4. Sentence 2, as written, infers that   emotional distress, fatigue, and premature menopause are the principal   mediators or factors associated with body/joint pain, etc. Is this really so?

We modified the introduction and made the aims and   necessity of the study more clear

Page 3a.   1st sentence. Text indicates that SF-36 has 8 subscales;   only 7 are listed. It is noted here that the rating scale ranges from 0   to100; the higher the rating relates to a lower or poorer QOL.

We added one other subscale and explained the   following Each subscale covers 2-10 questions and has a score from 0-100. Zero   is the lowest value and 100 is the highest. If the score is low, this points   to an impaired quality of life or impaired quality in the domain described by   the subscale (e.g. physical role, mental health etc).”

b.  Data analysis.(1)   provide an explanation why post-hoc groups   were categorized based also on hematocrit and WBC. (2)   why was mean age used instead of median age? (3)   reviewer perspective – the sample population   appears to be disproportionately tilted toward relatively young   (premenopausal) women. Is the age of most new diagnoses of breast cancer in   Greece different from other European countries and North America which is   predominantly postmenopausal?

This was done to assess whether quality of life was   different between normal range and abnormal range of hemoglobin, hematocrit   and White Cells Count. To the best of our knowledge there are not studies examined the association between anemia and   quality of life in breast cancer survivors

(2) The mean age was calculated on raw age data and   not on the age categories. Given that the distribution of age was normal   (Skewness: .170, SE= .306, z= 0.55,   Kurtosis: -0.712, SE = 0.604, z =   1.18), the mean and standard deviation could be used. The median age was 50.

(3) We found the sample of two other studies in   Greece had a similar mean age in breast cancer patients to our sample.

These studies are:

1)             Fradelos   EC, Latsou D,Mitsi D, Tsaras K, Lekka D, Lavdaniti M, Tzavella F,   Papathanasiou IV. Assessment of the relation between religiosity, mental health, and psychological   resilience in breast   cancer patients.   Contemp Oncol (Pozn) 2018; 22(3):   172–177. 

2)         Patsou   ED, Alexias GT, Anagnostopoulos FG, Karamouzis MV. Physical   activity and sociodemographic variables related to global health, quality of life, and psychological factors in breast cancer survivors.Psychol Res Behav   Manag. 2018; 11:   371–381

Page 4

a.  Table 2, treatment type. Did any patient get   post-surgical radiation?

b.    Table 3. SDs are very wide.

c.     Specifically how should the statement “Age had   a significant effect on the role physical…” be interpreted?

d.    Table 4. According to the data presented,   older patients (i.e., >60 years of age) had better role   physical ratings than the other age groups. Was part of the explanation for   this finding explained by extent of surgery (i.e., total mastectomy) thus   foregoing radiation?

a.                 The patients who collected   in our sample was not undergoing radiotherapy  yet

b.     We calculated medians and   iqrs

c.      We changed the result and   we added the following Table 4 shows the effects of age   (categories) on the different quality of life domains. The effect of age on   the Role Physical was significant [(F(3) = 3.26, p =.03]. Post-hoc   comparisons revealed no significant differences across the age groups.   However, there was a marginal difference between the 29-39 and the 60+ groups   (p = .053) showing that the younger participants possibly had a better   quality of life regarding the Role Physical than participants that were over   60 years old. There were no other differences across the age group on the   other SF-36 subscales (ps > .1) age categories on quality of life and its   domains

d.                We modified the   statistical analysis and we found that “Post-hoc comparisons   revealed no significant differences across the age groups. However, there was   a marginal difference between the 29-39 and   the 60+ groups (p = .053) showing that the younger participants possibly had   a better quality of life regarding the Role Physical than participants that   were over 60 years old. There were no other differences across the age group   on the other SF-36 subscales (ps > .1)

Page 6.

a. Discussion. Frequently mentioned is the use   of the terms “low score” which was initially defined in the Instrument   section (i.e., lower score being associated with better QOL). In this section   the use of these two words appears to be associated with poorer QOL. Please   reconcile this discrepancy. Paragraph 3 is another example of this confusing   message.

b. Paragraph 8, 1st sentence. What exactly is being described here?

 a. The   discussion is rewritten and we corrected the mistake about the scores of   SF-36.

B) We discuss almost all the new results, so the   sentence is changed with another

Reviewer 2 Report

The manuscript  is interesting for  the  subject  investigated . Quality of life in cancer is one of the most important aspect in metabolic chronic disease and the  evaluation of  the potential variables  involved in the modifications of  this  context is the principal originality of this investigation . 

 However  the introduction, results  and  discussion need  to be implemented  in terms  of the  evaluation of  the physical  activity practiced on behalf of  the women studied .   The sports  activity or moderate physical activity  of  the past  and  in the  background  of  the  group analyzed   could be important to better evaluate  the data obtained . Some  others considerations of the  clinical implications and impact  of these kind  of approach need to be considered . Do  you  think that  this kind of investigation needs  to   be spread  to  all the oncological centers ?

Author Response

Factors influencing the quality of life in breast cancer patients six months after the completion of chemotherapy

The manuscript is interesting for the subject  investigated . Quality of life in cancer is one of the most important aspect in metabolic chronic disease and the evaluation of  the potential variables  involved in the modifications of  this  context is the principal originality of this investigation . 

 However  the introduction, results  and  discussion need  to be implemented  in terms  of the  evaluation of  the physical  activity practiced on behalf of  the women studied .   The sports  activity or moderate physical activity  of  the past  and  in the  background  of  the  group analyzed   could be important to better evaluate  the data obtained . Some others considerations of the clinical implications and impact  of these kind  of approach need to be considered . Do  you  think that  this kind of investigation needs  to   be spread  to  all the oncological centers ?

Response: We modified statistical analysis and we found and other results about the other domains of quality of life e.g. emotional role, social functioning, bodily pain, vitality, etc. Also,, in the discussion we discuss these results about the aforementioned domains.

It is important for breast cancer survivors to have moderate physical activity but this is another study, and could be done and spread in another oncological centers. In this study we examined quality of life and its domain and we referred in physical function as a domain of quality of life.

Reviewer 3 Report

The introduction highlights the fact that breast cancer affects quality of life (QoL) in a variety of ways. Clearly illness will affect QoL.  A list of factors found in other studies is given but there is limited critical analysis of these findings. No specific research issues emerge from the introduction.

The only rationale is that there are few Greek studies on this topic.

Line 33-34. The subjects of the study are patients who have completed six months chemotherapy. However, the paper does not state what their health was like-isn't this a major factor that has to be considered?

Data Analysis

Line 98-100. To enable statistical analysis of the clinical data, participants were assigned post-hoc to different groups. Could you explain this?.

It is not really clear how the data analysis section relates to the study objectives. There could be a clearer list of factors and the rationale for their inclusion.

Also the analysis should be multivariate as the factors affecting QoL are likely to be correlated. Also there could be interactions relating to QoL measures, eg cancer stage x age.

Results

124. There should be more discussion of table 3. Why does the paper highlight variables with lower means? Isn’t it the higher scores that indicate more impairment. (E.g. bodily pain=80.93)?

There could be more discussion of which QoL variables are best predicted by the independent variables?   The finding that only physical QoL measures are significant could be further discussed.

Perhaps there should be a table summarizing which outcomes are/are not affect would be useful.,. 

Looking at tables 1 and 2, is it the case the low numbers in many of the cells indicate non-significance (even when mean values are different)?

144-151. Multiple reports of non-significance could be due to sample size.

Results. There are limitation in the reporting of the results due to sample size and the type of analysis. Only physical outcomes are reported. As noted above,  what about all the other outcome domains, eg mental health, social functioning etc?

159.  I don’t understand this statement: the mean scores of the SF-36 subscales varied between 42 and 80 implying that breast cancer survivors experience acceptable levels of quality of life. First, there is wide variability; second, no reference data are cited; third what does “acceptable mean”.

161. Previous comment is  contradicted by statement that “breast cancer patients experience low scores in the physical functioning subscale.” But not related to duration of illness etc?

P177. Another interesting finding is that “participants experience low scores of emotional role and mental health”. Compared to who? 

See comments above-but what factors (other than the illness)  does this relate to?

Overall there may be potentially interesting results but the currently analysis sheds little light on the factors associated with quality of life among women experiencing treatment for breast cancer. 

The main results seem fairly obvious.

There should be a clear statement of the study limitations. There should be a clear description of the analysis strategy and how this relates to specific research questions. 

There is no consideration of the implications of the findings, beyond knowing that breast cancer impacts on some aspects of QoL. 

What of those outcomes where no significant associations were found etc.

Author Response

Reviewer   3

ANSWERS

The introduction   highlights the fact that breast cancer affects quality of life (QoL) in a   variety of ways. Clearly illness will affect QoL.  A list of factors   found in other studies is given but there is limited critical analysis of   these findings. No specific research issues emerge from the introduction.

The only   rationale is that there are few Greek studies on this topic.

We changed the introduction and we stressed the   results of other studies about the factors influence quality of life. We   tried to stress the necessity of our study

Line 33-34. The subjects of the study are   patients who have completed six months chemotherapy. However, the paper does   not state what their health was like-isn't this a major factor that has to be   considered?

We added   the following sentence  “Their   performance status was assessed with ECOG which is 0” which shows that   patients were fully active

Line 98-100. To enable   statistical analysis of the clinical data, participants were assigned   post-hoc to different groups. Could you explain this?.

It is not really   clear how the data analysis section relates to the study objectives. There   could be a clearer list of factors and the rationale for their inclusion.

Also the   analysis should be multivariate as the factors affecting QoL are likely to be   correlated. Also there could be interactions relating to QoL measures, eg cancer   stage x age.

Line 98-100. Hemoglobin, hematocrit and white cell   count data were categorized to different groups according to whether their   levels were low, normal or higher than normal. As in these measures, the   higher or the lower does not mean the better, correlations between these   measures and sf-36 scores could not be informative.  So assigning post-hoc participants to   groups according to the levels of these measures allowed analysis. Given   this, we examined whether SF-36 scores differed across this categories.

The citation of the normal range of red and white blood cells is   referred in the text

Also we use these clinical and demographic   characterizes because there are conflicting    results how they influence the domains of quality of life

124. There   should be more discussion of table 3. Why does the paper highlight variables   with lower means? Isn’t it the higher scores that indicate more impairment?   (E.g. bodily pain=80.93)?

There could be   more discussion of which QoL variables are best predicted by the independent   variables?   The finding that only physical QoL measures are   significant could be further discussed.

Perhaps there   should be a table summarizing which outcomes are/are not affect would be   useful.,. 

Looking at   tables 1 and 2, is it the case the low numbers in many of the cells indicate   non-significance (even when mean values are different)?

We corrected the description of the scale as   mentioned above.

In new statistical analysis we made new tables in   which we described the associations between clinical and demographic   characteristics and the domains of quality of life.

144-151. Multiple reports of non-significance   could be due to sample size.

Results. There are limitation in the reporting   of the results due to sample size and the type of analysis. Only physical   outcomes are reported. As noted above, what about all the other outcome   domains, eg mental health, social functioning etc?

We modified the discussion based on new results.  We explained the new results based on   findings of other studies.  We discuss   and another results about mental health and social functioning

159.  I   don’t understand this statement: the mean scores of the SF-36 subscales   varied between 42 and 80 implying that breast cancer survivors experience   acceptable levels of quality of life. First, there is wide variability;   second, no reference data are cited; third what does “acceptable mean”.

We omitted this   sentence and we replaced with the following sentence “Breast   cancer patients in the study sample experienced low level of physical   functioning, mental health, emotional role and vitality. This is in   consistent with other studies [21,26].”

161. Previous   comment is contradicted by statement that   “breast cancer patients experience low scores in   the physical functioning subscale.” But not related to duration of   illness etc?

We omitted this sentence and we replaced with   discussion of the results.

P177. Another interesting finding is that   “participants experience low scores of emotional role and mental health”.   Compared to who? 

We omitted this sentence and we replaced with   discussion of the results.

There should be   a clear statement of the study limitations. There should be a clear   description of the analysis strategy and how this relates to specific   research questions. 

We changed study   limitations with the following
  “This study has some limitations. It was conducted in a one hospital   located in a major Greek city; the sample is relatively small, so that the   results cannot be generalized with respect to the entire Greek population. Another   limitation had to do with the cross-sectional nature of the present study.  We could not assess the trajectory of   quality of life at the end the chemotherapy, six months on, one year or five   years on. A future study in Greece employing a longitudinal design   could provide more clear conclusions. Although we study the influence of   demographic and clinical characteristics the present study did   not investigate some other important clinical characteristics such as type of   chemotherapy regimen, the duration of it etc. However, the results provide   valuable information for the issue at hand, and illustrate the great need for   further longitudinal studies in order to draw reliable conclusions. Despite these limitations our study has one significant strength. To our   knowledge, this is the first population-based study to investigate factors   influence quality of life in breast cancer survivors women in Greece, where   culture, lifestyle are significantly different from those in Western   populations.”

There is no   consideration of the implications of the findings, beyond knowing that breast   cancer impacts on some aspects of QoL. 

We added some   implications in the conclusions “Health care professionals need to be   more aware of quality of life issues within this group of cancer patients in   order to meet their needs, following acute treatment but also for the   duration of their struggle with this disease. Also they should arrange care   plans, take into account these factors, in order to find ways to increase   patients quality of life.”

Round 2

Reviewer 3 Report

Some changes have been made to the text, although not all the points I made have been addressed. 

For example I noted that: "There could be   more discussion of which QoL variables are best predicted by the independent   variables?   The finding that only physical QoL measures are   significant could be further discussed." This point is not fully addressed.

There could still be greater clarity in the presentation of results and discussion. 

Because the discussion covers a wide range of factors associated with quality of life in breast cancer patients, I found it difficult to follow. By the end of the paper I was unclear as to what factors could be addressed by health care professionals.  I would suggest being more precise in the abstract which currently reads. "Health care professionals should be more aware of these quality of life aspects within this group of cancer patients in order to meet their needs following acute treatment".

Also in the discussion: Health care professionals need to be   more aware of quality of life issues within this group of cancer patients in   order to meet their needs, following acute treatment but also for the   duration of their struggle with this disease. Also they should arrange care   plans, take into account these factors, in order to find ways to increase   patients quality of life.”

Again in what way can the arrange care plans to take account of these factors-give an example.

Author Response

Some changes have been made to the text, although not all the points I made have been addressed. 

For example I noted that: "There could be   more discussion of which QoL variables are best predicted by the independent   variables?   The finding that only physical QoL measures are   significant could be further discussed." This point is not fully addressed.

RESPONSE we added in the discussion the following

It is worthwhile to note that we did not find any influence of demographic on mental health domain of quality of life. This is in accordance with the findings of other studies [9,20].There is a need for further research in Greece to clarify if the aforementioned factors influence mental health and should be used more specific questionnaires.

Also, in the discussion we referred the influence of type of treatment on mental health 

There could still be greater clarity in the presentation of results and discussion. 

RESPONSE We have done now changes in the results section. However, it was not very clear what was meant regarding clarity in the presentation of the results. Which part are not clear enough? 

Because the discussion covers a wide range of factors associated with quality of life in breast cancer patients, I found it difficult to follow. By the end of the paper I was unclear as to what factors could be addressed by health care professionals. 

RESPONSE we added in the discussion the following

 In this study we found that age, menopausal status and hormone therapy influenced quality of life in breast cancer patients.

 I would suggest being more precise in the abstract which currently reads. "Health care professionals should be more aware of these quality of life aspects within this group of cancer patients in order to meet their needs following acute treatment".

RESPONSE   We changed the sentence with the following sentence:

Health care professionals should be more aware of the factors that influenced the quality of life domains (role physical, bodily pain and vitality) within this group of cancer patients in order to meet their needs following acute treatment.

Also in the discussion: Health care professionals need to be  more aware of the quality of life issues within this group of cancer patients in   order to meet their needs, following acute treatment but also for the   duration of their struggle with this disease. Also they should arrange care   plans, take into account these factors, in order to find ways to increase   patients quality of life.”

Again in what way can the arrange care plans to take account of these factors-give an example

RESPONSE

We modified the sentence in the conclusions as

 Also they should arrange care plans, take into account these factors, in order to find ways (not pharmacological interventions) to increase patients quality of life.